# Discovery and Genomic Characterization of a Novel Hepadnavirus from Asymptomatic Anadromous Alewife (*Alosa pseudoharengus*)

**DOI:** 10.3390/v16060824

**Published:** 2024-05-22

**Authors:** Clayton Raines, Jan Lovy, Nicolas Phelps, Sunil Mor, Terry Fei Fan Ng, Luke Iwanowicz

**Affiliations:** 1U.S. Geological Survey, Eastern Ecological Science Center, Leetown Research Laboratory, 11649 Leetown Road, Kearneysville, WV 25430, USA; craines@usgs.gov; 2West Virginia Cooperative Fish and Wildlife Research Unit, Davis College of Agriculture, Natural Resources & Design, West Virginia University, 1 Waterfront Pl, Morgantown, WV 26506, USA; 3U.S. Geological Survey, Western Fisheries Research Center, Seattle, WA 98115, USA; jlovy@usgs.gov; 4NJ Fish & Wildlife, Office of Fish and Wildlife Health and Forensics, 605 Pequest Rd, Oxford, NJ 07863, USA; 5Department of Fisheries, Wildlife, and Conservation Biology, College of Food, Agriculture, and Natural Resource Sciences, University of Minnesota, St. Paul, MN 55108, USA; phelp083@umn.edu; 6Animal Disease Research and Diagnostic Laboratory, South Dakota State University, 1155 North Campus Drive, Brookings, SD 570077, USA; sunil.mor@sdstate.edu; 7Veterinary Diagnostic Laboratory, Department of Veterinary Population Medicine, University of Minnesota, Saint Paul, MN 55455, USA; 8Department of Pathology, University of Georgia, Athens, GA 30602, USA; terryng@uga.edu; 9U.S. Department of Agriculture-Agricultural Research Service, National Center for Cool and Cold Water Aquaculture, Kearneysville, WV 25430, USA

**Keywords:** hepatitis B virus, *Hepadnaviridae*, fish virus, alewife

## Abstract

The alewife (*Alosa pseudoharengus)* is an anadromous herring that inhabits waters of northeastern North America. This prey species is a critical forage for piscivorous birds, mammals, and fishes in estuarine and oceanic ecosystems. During a discovery project tailored to identify potentially emerging pathogens of this species, we obtained the full genome of a novel hepadnavirus (ApHBV) from clinically normal alewives collected from the Maurice River, Great Egg Harbor River, and Delaware River in New Jersey, USA during 2015–2018. This previously undescribed hepadnavirus contained a circular DNA genome of 3146 nucleotides. Phylogenetic analysis of the polymerase protein placed this virus in the clade of metahepadnaviruses (family: *Hepadnaviridae*; genus: *Metahepadnavirus*). There was no evidence of pathology in the internal organs of infected fish and virions were not observed in liver tissues by electron microscopy. We developed a Taqman-based quantitative (qPCR) assay and screened 182 individuals collected between 2015 and 2018 and detected additional qPCR positives (n = 6). An additional complete genome was obtained in 2018 and it has 99.4% genome nucleotide identity to the first virus. Single-nucleotide polymorphisms were observed between the two genomes, including 7/9 and 12/8 synonymous vs nonsynonymous mutations across the polymerase and surface proteins, respectively. While there was no evidence that this virus was associated with disease in this species, alewives are migratory interjurisdictional fishes of management concern. Identification of microbial agents using de novo sequencing and other advanced technologies is a critical aspect of understanding disease ecology for informed population management.

## 1. Introduction

The collective term “river herring” refers to the anadromous iteroparous alosine filter feeders blueback herring (*Alosa aestivalis*) and alewife (*A. pseudoharengus*). River herring have long been utilized as an easily harvested and inexpensive source of food, bait, and fertilizer. Specific use of river herring is documented as early as the arrival of the pilgrims to what is now the United States [1,2], and substantially earlier by indigenous peoples [3]. Consequently, the availability of river herring has significant economic impacts for the communities around the fishery [1,4]. Reductions in abundance were often met with reintroduction efforts and the translocation of adult fish to re-establish populations [5,6]. As with other anadromous fishes, adults migrate from marine environments into freshwater for spawning. For river herring, this occurs predominantly in eastern North American streams, lakes, and ponds, depending on environmental conditions and food availability [6,7,8]. Perhaps as a direct result of their social and economic value and habitat loss due to damming, anadromous river herring stocks have declined significantly over the past 400 years, and are now considered species of concern by the National Marine Fisheries Service (National Oceanic and Atmospheric Administration [NOAA]) [9,10] and the Atlantic States Marine Fisheries Commission [11]. However, predation, habitat quality, climate change, and disease constitute additional threats which can indirectly or directly cause population declines [12,13]. Understanding and incorporating management approaches to mitigate fish health threats to these species is critical to long-term health and population sustainability.

Hepadnaviruses are enveloped, reverse-transcribing DNA viruses of the family *Hepadnaviridae*. While this family was historically represented by the *Avihepadnavirus* and *Orthohepadnavirus*, de novo sequencing methods have revealed a greater diversity of other genera that include the *Herpetohepadnavirus*, *Metahepadnavirus*, and *Parahepadnavirus* [14]. At present, it appears that these genera are restricted to piscine or herptile hosts [14]. The avi- and orthohepadnaviruses are hepatotropic and exhibit a narrow host range. Infections are transient or chronic, and in the latter case, *Orthohepadnavirus* infections are associated with an increased risk of hepatocellular carcinoma. Much less is known about the tissue tropism of these more recently described hepadnaviruses, but there is evidence that they may not exhibit the same tissue restrictions [15,16]. Similarly, the disease ecology and significance in respect to host health are uncharacterized. 

The white sucker hepatitis B virus (WSHBV) is a parahepadnavirus and is perhaps the best studied of the three newly recognized genera. This virus, discovered from white suckers (*Catostomus commersonii*), was the first reported hepadnavirus to infect fish, and phylogeographic analyses suggest that genomic diversity occurs similarly to that of well-studied hepadnaviruses [17]. While liver and skin tumors are often observed in white suckers in regions of the Great Lakes where chemical contamination and WSHBV are reported, there are no clear associations between viral infection and neoplasia [18,19,20]. It is notable that the metahepadnavirus (bluegill hepatitis B virus) was observed in a fish with an observed tumor, but, like the scenario with the WSHBV, datasets are not available to ascribe or even suggest a causative association with neoplastic disease [15]. In addition, partial genomes of fish hepadnaviruses of unknown pathogenicity have been identified in African cichlids and a number of marine fish hosts via metagenomic surveys [16,21,22]. Here, we identify a novel hepadnavirus that is most related to other described metahepadnaviruses in clinically normal migratory alewives sampled from the Great Egg Harbor and Maurice Rivers in New Jersey, USA. As alewives are a species of concern, investigation into possible pathogens is essential to support their long-term health and population sustainability.

## 2. Materials and Methods

### 2.1. Sampling

Three life stages of anadromous river herring from New Jersey, USA were examined opportunistically in this study, including spawning adult fish and young-of-the-year (age-0) fish from the river environment. A population of landlocked alewives were also included in the study to determine if viruses were detected in these isolated fish populations, which have adapted to complete their entire life cycle within a freshwater lake.

#### 2.1.1. Spawning River Herring

Spawning adult anadromous alewives and blueback herring were collected in March and April 2015, 2016, and 2018 during the New Jersey Division of Fish and Wildlife (NJDFW) annual surveys (Figure 1). Fish were captured by gill net, set for 2 h at a time, from the Maurice River and the Great Egg Harbor River (Table 1). Following transport to shore, the fish were euthanized with an overdose of MS-222 and transported on ice to the Pequest Aquatic Animal Health Laboratory (Oxford, NJ, USA). Fish were either sampled fresh or immediately frozen for future sample collection; sample collection information is summarized in Table 1. 

In 2015, spleen, kidney, brain, and gills were aseptically dissected from the 14 fish, and up to 3 fish were pooled together and processed for viral cell culture assays, further described below. In 2016 and 2018, head, kidney, spleen, and liver were aseptically removed and transferred to 2 mL Eppendorf tubes and frozen at −80 °C until shipping to the Minnesota Veterinary Diagnostic Laboratory for cell culture assays or genetic analysis (Table 2). The internal organs, including liver, heart, anterior and posterior kidney, spleen, and gastrointestinal tract, were preserved in 10% neutral-buffered formalin for histology.

#### 2.1.2. Young-of-the-Year (Age-0) River Herring

Young-of-the-year river herring were collected in 2015 during the NJDFW annual survey from the Great Egg Harbor and Delaware Rivers (Figure 1). Fish from the Great Egg Harbor River were collected via mesh beach seine, whereas those from the Delaware River were collected by boat electrofishing. Collected fish were euthanized with an overdose of MS-222 and transported on ice to the Pequest Aquatic Animal Health laboratory for necropsy and tissue sample collection (Table 2). Spleen, kidney, gills, and brain were aseptically removed for processing by viral cell culture assays. Subsets of combined viscera were preserved for genetic analysis. 

#### 2.1.3. Landlocked Alewife

In 2015, landlocked alewives were collected in collaboration with a commercial bait shop from Lake Hopatcong located in northern New Jersey, USA (Figure 1). Fish had been recently collected at night, using lights to attract the fish. An umbrella net was deployed off a barge to capture the fish. The fish were transported to shore and held in flow-through tanks supplied with lake water for up to 72 h. Fish were transferred alive to the Pequest Aquatic Animal Health Laboratory, where the fish were euthanized with an overdose of buffered MS-222 and spleen, kidney, brain, and gills were aseptically collected for viral cell culture assays. Subsets of viscera homogenate were preserved for genetic analysis (Table 2). 

### 2.2. Viral Screening (Cell Culture)

Spawning adult and Age 0 anadromous river herring from 2015 were screened for viruses using viral cell culture assays run at the Animal Health Diagnostic Laboratory, New Jersey Department of Agriculture (Ewing, NJ, USA). Tissue samples were processed by standard methods. Briefly, the tissue pools were homogenized in Hanks’ Balanced Salt solution (HBSS) using a stomacher, suspended, incubated in antibiotic media overnight, and centrifuged to purify viral agents within the supernatant. The supernatant was inoculated onto two cell lines, the epithelioma papulosum cyprini (EPC) cell line incubated at 15 and 20 °C, and the Chinook salmon Embryo-214 (CHSE-214) cell line incubated at 15 °C. Duplicate wells within 24-well plates were inoculated for each cell line/incubation temperature. The cells were evaluated 3 times per week for two weeks, after which, if no cytopathic effects (CPEs) were noted, samples were blind passaged by re-inoculation on fresh cells for an additional two weeks. If no CPEs were noted 2 weeks after blind passage, then the samples were reported negative. The remaining supernatant that was unused for viral cell culture assays was frozen at −80 °C for further genetic analysis.

### 2.3. DNA Extractions and Template Preparation

Subsampled organs were preserved in RNAlater™ and DNA was extracted for High-Throughout Sequencing (HTS), Sanger sequencing, or quantitative PCR (Table 2). Samples were stored at room temperature for 24 h prior to being stored at −20 °C until extraction. All DNA was extracted using a DNeasy Blood and Tissue Kit (Qiagen, CA, USA), following manufacturer instructions.

### 2.4. High Throughput Sequencing-Assisted Virus Discovery

Extracted DNA from samples collected in 2015 was submitted to the University of Minnesota Genomics Centre (UMGC) for library preparation. An Illumina Nextera XT Library Preparation Kit (Illumina, CA, USA) was used in accordance with the Nextera XT Library Preparation Reference Guide (Doc# 15031942 ver. 5) for HiSeq preparation. Prepared libraries were diluted per standard protocols within the HiSeq System Denature and Dilute Libraries Guide (Doc # 15050107 ver. 06). Pooled indexed libraries were then run for 2 × 151 cycles. 

### 2.5. Bioinformatic Discovery of a Novel Hepadnavirus

A draft linear genome was assembled from a pooled adult alewife sample collected from the Maurice River in 2015 and presumptively identified as a novel hepadnavirus. PCR primers were designed from this genome and non-pooled samples were screened to identify PCR-positive individuals. Additionally, to increase sensitivity, qPCR primers and probes were designed to facilitate higher sampling throughput. DNA was isolated from an individual PCR-positive alewife adult (M10) to establish a complete genome of this virus. We used rolling circle amplification (RCA) to enrich circular DNA templates for increased sequencing coverage of the viral genome. In addition, we used this method to sequence an additional genome from a positive adult alewife (MR4) collected from the Maurice River in April 2018 identified via qPCR. RCA was initiated by combining template viral DNA, 500 µM Exo-resistant primers, and 1× Equiphi reaction buffer into a cocktail. The cocktail was heat denatured for 3 min at 95 °C and then immediately cooled on ice for 5 min. The cooled, denatured cocktail was then combined with a final reaction mix of 10× Equiphi reaction buffer, 100 µM DTT (dithiothreitol), dNTPs, N.F. water, and Equiphi29 polymerase (ThermoFisher Scientific, Waltham, MA, USA) and incubated at 45 °C for 3 h and 65 °C for 10 min before being immediately frozen and stored at −80 °C until sequencing. RCA products were then quantified with a Qubit 4.0 Fluorometer (Invitrogen, Carlsbad, CA, USA) and normalized as starting material for NGS library prep. An Illumina Nextera XT Library Preparation Kit (Illumina, San Diego, CA, USA) was used in accordance with the Nextera XT Library Preparation Reference Guide (Doc # 15031942 ver. 5) for MiSeq preparation. Normalization of the final library was performed with Illumina’s Bead-Based Normalization (BBN) method and pooled as described in the BBN Loading Concentrations Exceptions Table 2 identified in MiSeq System Denature and Dilute Libraries Guide (Doc # 15039740 ver. 10). Pooled libraries were then run for 2 × 301 cycles and loaded with a 15% PhiX (12.5 pM) spike. In addition, primers which spanned areas of lower coverage (Primer ID 1–4; Table 3) were implemented to enhance HTS methods.

After sequencing, paired reads were quality screened and trimmed prior to de novo assembly using CLC Genomics Workbench version v.22.0.2 (https://digitalinsights.qiagen.com). Once complete genomes of the alewife hepadnavirus (ApHBV) were constructed, we predicted open reading frames (ORF) and identified repeat regions using Geneious Prime v.2022.2.2 (http://www.geneious.com). Protein characteristic analyses, including isoelectric point estimates, were also conducted using Geneious Prime. All predicted proteins were queried against the PDB, Pfam-A, UniProt-SwissProt-viral70, and NCBI Conserved Domains databases via HHpred [23]. We analyzed the two alewife hepadnavirus genomes for synonymous and nonsynonymous bases using DnaSP (v.6) [24].

### 2.6. Quantitative PCR

We designed a TaqMan qPCR assay to facilitate screening of the virus (Primer ID 5–7; Table 3) and determine prevalence. Primers were designed using Primer3 v2.3.7 [25], bundled in Geneious Prime. We targeted the hepatitis core conserved domain locus from sample M10. Standards were created from a synthesized 150 bp gBlock^TM^ gene fragment (Integrated DNA technologies Inc., Coralville, IA, USA) run in a 10-fold serial dilution from 1–1 × 10^8^ copies. The reactions were run on a QuantStudio 5 (ThermoFisher, Waltham, MA, USA) with default settings. We confirmed a single 117 bp product via end-point PCR and agarose electrophoresis. In addition, melt curve analysis was performed using the TaqMan forward and reverse primers in a SYBR-based format. That assay used the TaqMan cycling conditions and profile in Table 3, but without the TaqMan probe, and SYBR^TM^ Green (Applied Biosystems, Carlsbad, CA, USA) to form a master mix. To confirm a single amplification peak, melt curve analysis was run with parameters of 95 °C for 15 s cooling at a rate of 1.6 °C/s, and then followed by 60 °C for 60 s and 95 °C for 15 s. A total of 32 pooled samples comprised 160 discrete specimens (5 fish per pool) and 22 individual fish samples were screened via qPCR (Table 2).

## 3. Results

### 3.1. Cell Culture Assays, Histology, and Transmission Electron Microscopy

No CPE was detected in any of the cell culture assays. Histopathologic evaluation did not detect lesions suggestive of virus infection and transmission electron microscopy revealed no hepadnavirus-like particles or any other observable virus particles. Histopathology detected notable myxozoan parasites infecting the kidney, which was the focus of a separate study [26]. 

### 3.2. Virus Prevalence in Wild Caught Herring

The qPCR assay developed here targeted a 117 bp region of the core protein coding region and was effective for the identification of virus-positive individuals. The R^2^ of the standard curve was 0.994 and the efficiency of the reaction was 96.3%. The limit of detection was calculated to be nine copies. Background amplification was observed in virus-negative samples, as determined by the absence of viral sequences obtained during sequencing. Given that there were no reliable means to interpolate a LoQ via CV of target amounts, we consequently set a diagnostic threshold for positive samples >10 copies, utilizing our lowest standard, consistent with a CV ≤ 35% [27]. Of the 32 pooled samples screened, six were positive via qPCR (Figure 2). ApHBV was exclusively detected in adult, migratory alewives collected in freshwater, riverine environments (31.6% positive) during the spring spawning run. Virus genome copies of virus-positive fish ranged from 15 copies to 10,228,303 copies. The median copy number was 213. Virus-positive individuals were sourced exclusively from the Great Egg Harbor River in April of 2018 (n = 1) and Maurice River in 2015 (n = 5). 

### 3.3. Sequencing the Viral Genome and ORF Organization

The initial assembly of pooled fish yielded a 3112 bp linear genomic contig. The complete genome of the alewife metahepadnavirus, sequenced using RCA enrichment (GenBank accession numbers OQ859058 (isolate M10) and OQ859059 (isolate MR4)), was 3146 bp (Figure 3). Genome coverage was 751 and 2786× for samples M10 and MR4, respectively. The genome size and organization were similar to that of other metahepadnaviruses. The GC content of the complete genome was 50.3%.

In silico translation identified five partially or completely overlapping reading frames (+1, +2, and +3). These corresponded to the core protein, polymerase protein, and surface protein of prototypical hepadnaviruses, in addition to a putative ORFY and ORFZ orthologous to ORFs in the bluegill hepadnavirus [15]. A hepatitis B X protein homologue common to orthohepadnaviruses was not present. The non-canonical polyadenylation signal (TATAAA) typical of hepadnaviruses was not present in the genomic sequence. Additionally, we identified a pair of 12 nt direct repeat (DR) regions, TGTTACACAGGA, located within the putative ORFZ and the core ORF that spanned 199 nt. Similar to other fish hepadnaviruses, a non-protein coding region was predated between the putative ORF Z and ORF Core [15].

Reading frame 1 (RF + 1) encodes the core polyprotein (nt 106–669, 564 bp) of 188 amino acids (aa) with a theoretical average molecular weight of 21041.97 daltons. The size of this ORF was in the range of other hepadnaviruses but was more similar (in size) to that of the orthohepadnaviruses. A putative ORFZ (nt 2965–3126; 162 bp) was identified that encodes a small 54 aa protein that overlaps the terminal end of the polymerase ORF. While data are unavailable to validate the expression of this protein, a 25 aa locus similar to frog virus-3 uncharacterized protein 086L (Q6GZT2) was identified in this ORF, providing evidence that this is a legitimate viral protein. We also identified the hepatitis core antigen conserved domain within this ORF (nt 402–573, 171 bp). The predicted isoelectric point of this polyprotein was 9.59 and was within the range reported for avian and mammalian hepadnaviruses (pI 9.34–10.12). 

Reading frame 2 (RF +2) encodes the viral polymerase protein (nt 542 –2992; 2451 bp) of 816 aa and was similar in size to other hepadnavirus P proteins (785–902 aa). We identified ORF Y in this reading which was completely overlapped by the core ORF. This ORF is in the same reading frame as the orthologous ORF Y of the bluegill hepadnavirus and the Crocodile icefish metahepadnavirus (IMDV). RF +2 also contained conserved domains that included viral DNA polymerases and reverse transcriptase (Figure 3). The overall MW of the P protein was 90,556.89 with a predicted pI of 9.21. Analysis of the predicted polymerase protein using the resistance module in the hepatitis B virus database [28] indicated that this virus is presumptively sensitive to HBV antiviral drugs that target reverse transcriptase [29]. This ORF partially overlaps the core protein ORF and completely overlaps the surface protein. 

Reading frame 3 (RF + 3) of WSHBV (RF +3; nt 1059–2201; 1143 bp) was homologous to the large surface protein of hepadnaviruses. We also identified a conserved domain for major surface antigen from hepadnavirus (vMSA) in this ORF (nt 1512–2183; Figure 3). 

### 3.4. Genomic Diversity

We compared the complete viral genomes sampled from two individual fish captured during different sampling events (2015 and 2018) within the Maurice River. Pairwise alignment of these 3146 bp genomes revealed 99.4% nucleotide sequence identity (20 nt differences). Single-nucleotide polymorphisms were restricted to the polymerase and surface protein ORFs (Figure 4) observed between the two genomes that included 7/9 and 12/8 synonymous vs nonsynonymous mutations across the polymerase and surface protein open reading frames, respectively. Differences between the P protein for nucleotide and protein sequence identity were 0.96% and 0.86%, respectively. Likewise, these differences were 1.4% and 1.8% for the surface ORF for nucleotide and protein sequences, respectively. 

### 3.5. Pairwise Comparisons and Phylogenetic Analysis of P Proteins

Comparative pairwise alignment of fish meta- and parahepadnavirus P proteins revealed 31.0–52.6% amino acid identity (Figure 5; Table 4). Phylogenetic analyses of the P protein to a subset of other described hepadnaviruses (Figure 6) support the assignment of the alewife hepadnavirus (ApHBV) to the genus *Metahepadnavirus* ([30]; metahepadnavirus). ApHBV was observed to be most similar to the TMDV (*Mexican tetra hepadnavirus*), AMDV (*Astatotilapia metahepadnavirus*), and Ectodini metahepadnaviruses (Figure 6). 

## 4. Discussion

Herein, a novel hepadnavirus was discovered in migratory alewife sampled from the Maurice and Great Egg Harbor Rivers, NJ, tentatively referred to here as alewife hepadnavirus (ApHBV). The closely related genomes (99.4% identity) being detected three years apart raises the possibility that it circulated endemically in the alewife population sampled. This novel virus groups most closely to the metahepadnaviruses [15,16,21]. Despite the recent global interest in viral diversity and ecology, much remains unknown about piscine hepadnaviruses, though rapid discovery is ongoing. A substantial uptick in viral discovery in fishes and other organisms in recent times has been the direct result of bioinformatic screening of short read archive databases from deep-sequencing investigations as well as purposeful metagenomic investigations designed to identify novel viruses [21,22,31,32]. Consequently, evaluations of whole genomes can be accomplished more quickly and effectively than ever, enabling apposite comparisons among specific viruses of interest.

Comparison of two complete ApHBV genomes identified 20 single-nucleotide polymorphism sites across the genomes (Figure 4). These differences led to protein coding changes in the surface (9 aa) and polymerase (8 aa) proteins, but not the core protein or predicted small ORFs. Although we did not obtain full genomes from all sampling locations, geographic variation among viral strains is possible, as was observed with WSHBV [17]. More targeted geographic sub-sampling would be necessary to discern all causes of genomic alterations. Furthermore, no conclusive phylogenetic hypothesis to the origins of hepadnaviruses exists, but the existence of synonymous HBVs among reptiles and amphibians suggests evolution driven by virus-host cospeciation [15,33,34]. Recent research suggests the potential for ancient divergence into distinct “Meta-Ortho+ and “Herpeto-Avi” lineages [35], which is not disputed by the inclusion of ApHBV among fish metahepadnaviruses (Figure 6). Two clades of the currently known fish-associated hepadnaviruses (metahepadnaviruses and parahepadnaviruses) essentially group orthogonally to one another, with KNDV-Lp-2 (killifish nackednavirus) as an outgroup to the parahepadnaviruses. On the whole, fish HBVs are genetically diverse, and many possess a pathophysiology permitting non-specific tropism, allowing the potential for more frequent host switching and mutation [16,32]. Given the divergence and diversity of previously undescribed hepadnaviruses, de novo sequencing approaches coupled with specific virus discovery workflows are an appropri-ate means of identifying this viral “dark matter”. Recent efforts successfully detected the presence of hepadnaviruses using samples of lower pharyngeal jaw tissue or gill [32], which might support additional methods of less invasive viral screening.

While the survey efforts here represent an initial screening of geographically limited river herring populations, it is notable that virus was only detected in riverine adult alewife samples migrating to freshwater spawning grounds. Virus-positive fish were detected during multiple years from discrete sample sites more than 30 km apart. As alosines typically shoal, migrate, and spawn as part of a collective metapopulation [36], the presence of disparate virus detections merits additional inquiry to better understand its prevalence across a greater geographical range and the ecology of this virus circulating within these important fish populations. Virus-positive fish were collected as far as 30.6 km upstream from the mouth of the saltwater Delaware Bay (Maurice River; Figure 1). The 31.5% prevalence of virus among adult riverine alewives is comparable to the detection rates observed for WSHBV [20]. 

An additional commonality between our findings and those reported for WSHBV is that the fish sampled were collected during migratory spawning runs. Similar observations have not been reported for other hepadnaviruses identified from fish hosts. A high viral genome copy number (10,228,303 copies) was observed in only one virus-positive individual in this study, as similarly reported in WHSBV-positive white sucker [17]. Notably, the fish with the greatest number of viral genome copies was a female alewife. The sex of all fishes in the current study was not determined and sex steroid concentrations were not measured, but during this life history stage androgens are elevated in both pre-spawn male and female fishes. Of note, hepatitis B virus replication is enhanced by androgen and the androgen receptor in mice, and males are more likely to develop HBV-associated hepatocellular carcinoma [37,38]. Though other positive samples were considerably lower (less than 1000 copies), the sample number and sex data availability of virus-positive fish in the current study are insufficient to assess sex-associated differences.

Detection of ApHBV among alewife, but not syntopic migratory blueback herring, strengthens suspicions of host specificity. However, the limited sample size of adult blueback herring in this study warrants caution when making comparisons between species or evaluating host switching. Even so, recent research suggests that adaptive radiation within a geographically isolated region is associated with rapid virus diversification and interspecific transmission in African cichlids [32]. Interrelatedness and pairwise similarities of observed cichlid metahepadnaviruses (Figure 5) suggest that the viruses were widely distributed and likely diversified along with their hosts in multiple radiation events [32]. Similar species level divergence and subsequent viral divergence might also be observed in the case of river herring metahepadnaviruses. Theoretically, however, limited genetic separation among closely related species could also suggest lowered adaptive barriers to host jumping. Both river herring exhibit identifiable regional groupings across regional river populations [39], and despite hybridization across syntopic populations, there is also evidence of genetic bottlenecking across alewife populations in the northern portion of their range [40]. Potentially, bottlenecked populations could be subject to reduced fitness, posing additional threats [41].

A lack of detections of this virus may suggest its absence in Lake Hopatcong (landlocked population), or perhaps migratory spawning adult alewives are predisposed to ApHBV infections compared to other species or life history stages. The ApHBV was also not detected in juvenile alewives, or from landlocked alewife populations, despite these samples representing 90% of total fish screened by qPCR. Physiological differences between life stages could account for a lack of observed cases of ApHBV within juvenile alewives, though the simplest explanation could be a simple lack of exposure as the adult fish out-migrate before the eggs hatch [42]. Alewives migrating upriver in freshwater face different stressors than in their characteristic marine environment [6,9,43,44,45,46]. In addition, physiological changes associated with reproduction lead to seasonal stressors associated with energy repartitioning, from somatic growth to gonad and gamete development. 

With respect to adult river herring, migratory behavior associated with reproduction is likely of even higher energetic cost than reaching sexual maturity [47,48], including changes to osmotic balance, development, behavior, and immune function [49]. This is particularly true when fish passage is inhibited with barriers [43,50,51,52]. Frequently, migration barriers are mitigated via fishways or fish ladders [53]. Yet, despite special considerations, it has been long understood that migrating fish still experience elevated stress and crowding when utilizing fishways [54,55]. Seasonal migration of fish introduces the opportunity for disease spread between and among different populations, which likely experience increased disease risk as a result [52,56]. Such disease transmission is not guaranteed, however, even when in a confined geographic location [57]. Other considerations include climate change-facilitated habitat degradation which reduces suitable habitat available for use by river herring [9,58]. Moreover, virulence may be substantially increased by elevated temperatures, while exceeding the optimal thermal regime may further increase alewife disease susceptibility [12,59].

Anadromous river herring stocks have declined by as much as 98% since the 1700s, due at least in part to historic overfishing and habitat loss from river damming [1,9,10]. Minimal study into potential viral pathogens or other potential diseases of anadromous river herring has been accomplished to date [13,60], and additional research is vital to improve conservation efforts. The discovery of the novel ApHBV, and other recent related discoveries [22,32], suggests there are likely to be a myriad of undiscovered hepadnaviruses within other species of fish, including other alosines and their potential hybrids. ApHBV was identified in clinically normal fish using Illumina-based sequencing of DNA in a diagnostic situation where standard tissue culture methods were not adequate for virus discovery. Further study is necessary to identify the geographical range of ApHBV and, more importantly, determine the biological significance of this virus [17,61]. While it is not immediately clear if this virus has a significant health impact, molecular method-based screening and proactive surveillance of this virus is possible even in the absence of overt pathology associated with infection. Advances in sequencing technologies have significantly increased our ability to discover novel viruses and other pathogens in diseased and clinically normal fish hosts. ApHBV provides yet another example of the enigmatic diversity of fish hepadnaviruses, and its association with an interjurisdictional species of economic and ecological importance emphasizes the necessity for continued hepadnaviral investigations. 

## Figures and Tables

**Figure 1 viruses-16-00824-f001:**
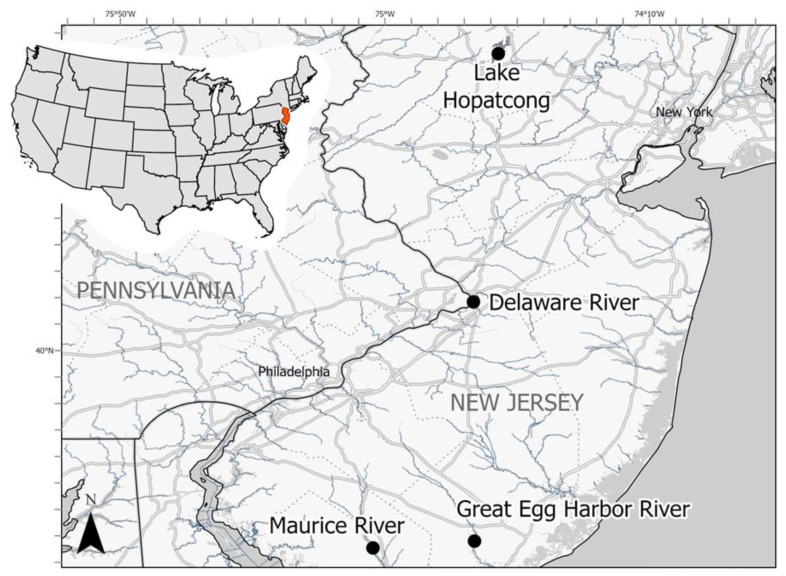
Collection locations of wild-caught alewife (*Alosa pseudoharengus)* and blueback herring (*A. aestivalis)* sampled between 2015 and 2018 in New Jersey, USA. Black dots correspond to locations of sampling sites. Red outline and fill denote the magnified location of New Jersey on the minimized map of the continental United States. All fish were collected live from freshwater environments. Base map from Esri and its licensors, copyright 2024.

**Figure 2 viruses-16-00824-f002:**
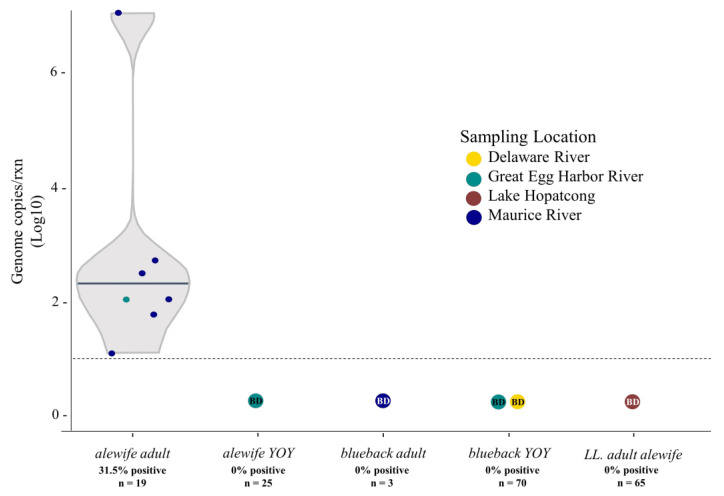
Violin plot of mean ApHBV genome copy numbers obtained from qPCR reactions associated with each sampled river herring life stage and sampling location. Gray horizontal line within the shape plot denotes the median genome copies/reaction. Dashed horizontal line designates the assay limit of quantification (LoQ; 10 copies). Labels within colored spheres identify sample locations screened where all samples screened were below the limit of detection (BD). Percentage designates the number of samples above the LOQ threshold compared to the total number screened, and (n) identifies the total number of samples screened of each life stage and species.

**Figure 3 viruses-16-00824-f003:**
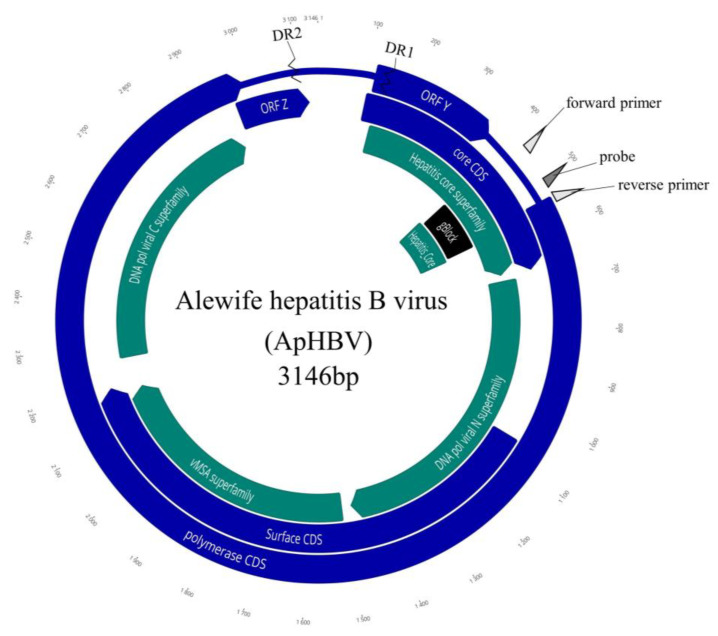
Genome organization of the alewife hepatitis B virus (ApHBV). The complete genome is 3146 nucleotides of dsDNA and includes three partially or completely overlapping ORFs encoding for the core, polymerase, and surface proteins, respectively (indigo), with highly conserved HBV coding sequences (teal). The synthesized 150 bp gene fragment gBlock is labeled in black. Primers and probes used in the qPCR assay are denoted by light gray and dark gray triangles, respectively.

**Figure 4 viruses-16-00824-f004:**
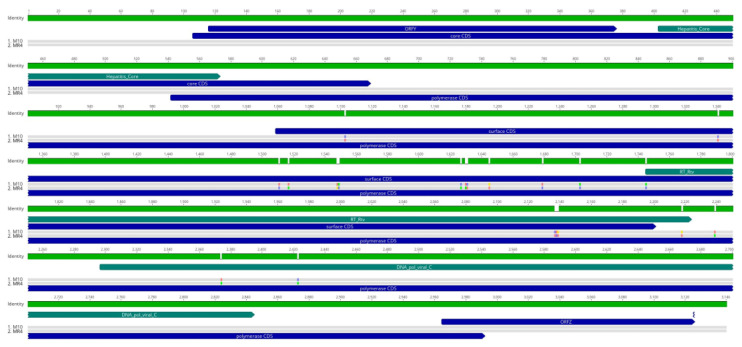
Genome alignments of the ApHBVs. Green bars denote nucleic acid identity, with breaks corresponding to single-nucleotide polymorphisms. Predicted protein coding sequence (indigo) and HBV conserved domains (teal) are graphically depicted. Gray bars identify obtained genomes, colored squares within gray bars denote nucleotide dissimilarities among obtained genomes.

**Figure 5 viruses-16-00824-f005:**
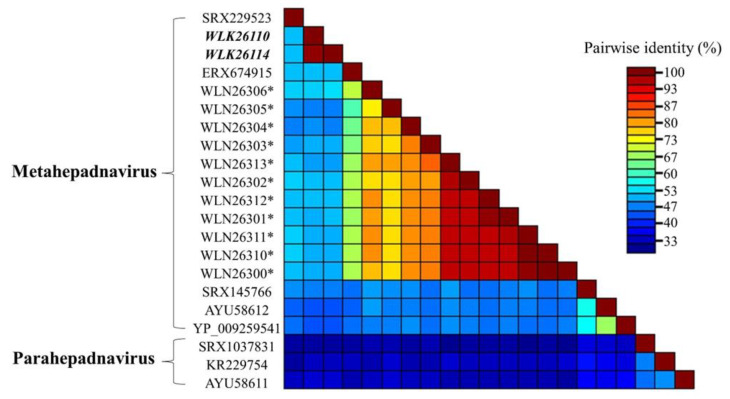
Pairwise sequence alignment identity (percentage) of hepadnavirus polymerase proteins based on pairwise alignment of fish Metahepadnaviruses and Parahepadnaviruses. Accession numbers for the novel alewife hepatitis B virus are labeled in bold and italics. Label names indicate NCBI accession numbers or NCBI SRA Experiments (prefix SRX/ERX). Asterisks denote metahepadnaviruses recently identified within Lake Tanganyika (African) cichlids.

**Figure 6 viruses-16-00824-f006:**
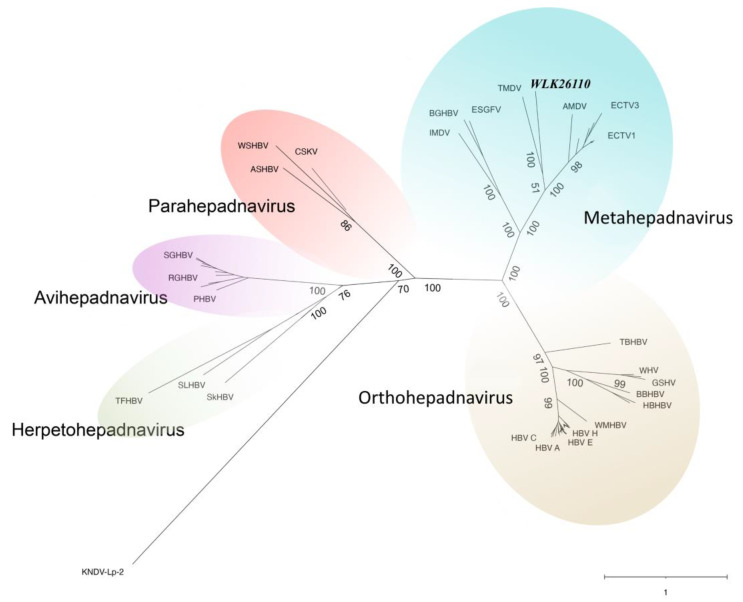
Radial phylogram depicting the relationships of the polymerase protein from 56 hepadnaviruses. Tip labels and branch support values were minimized for visual clarity. The novel alewife hepadnavirus (accession #WLK26110) is identified in bold italics and was most similar to TMDV (*Mexican tetra hepadnavirus*), AMDV (*Astatotilapia metahepadnavirus*), ECTV3, and ECTV1 (Ectodini metahepadnaviruses 3 and 1). Detailed information and metadata can be found in Appendix A.

**Table 1 viruses-16-00824-t001:** River herring collected from sampled New Jersey locations, including the Maurice River (Maurice), Great Egg Harbor River (GEH), Delaware River (Delaware) and Lake Hopatcong (Hopatcong). Abbreviations: Landlocked (LL), Total length (TL), Standard Deviation (SD), not determined (ND), SP adult (Spawning adult), YOY (young of year).

Site Name	Location	Species	Life Stage	Sample Date	Water Temperature	Collection Method	TL ± SD (mm)	Weight ± SD (g)	Sample Size
Maurice	39.378760, −75.037418	*Alewife*	Sp adult	April 2015	6.5–22 °C	3″ gill net	301 ± 10.5	252 ± 17.9	n = 12
Maurice	39.378760, −75.037418	*Blueback*	Sp adult	April 2015	6.5–22 °C	3″ gill net	297 ± 7.8	295 ± 21.6	n = 2
Maurice	39.378760, −75.037418	*Alewife*	Sp adult	March–April 2016	9.7–17 °C	3″ gill net	282 ± 10	ND	n = 16
Maurice	39.378760, −75.037418	*Alewife*	Sp adult	April–May 2018	5.2–13.8 °C	3″ gill net	277 ± 1.1	214 ± 26	n = 15
Maurice	39.378760, −75.037418	*Blueback*	Sp adult	April–May 2018	5.2–13.8 °C	3″ gill net	277 ± 0.3	199 ± 11	n = 3
GEH	39.400229, −74.717855	*Alewife*	Sp adult	April 2018	4.4–15 °C	3″ gill net	298 ± 1.3	246 ± 27	n = 4
GEH	39.400229, −74.717855	*Alewife*	YOY	August 2015	ND	100′ × 6′ × ¼″ mesh beach seine	61.5 ± 2.4	ND	n = 20
GEH	39.400229, −74.717855	*Blueback*	YOY	August 2015	ND	100′ × 6′ × ¼″ mesh beach seine	70.7 ± 4.3	ND	n = 40
Delaware	40.153580, −74.721212	*Blueback*	YOY	August 2015	ND	Boat Electrofishing	65.5 ± 4.2	ND	n = 60
Hopatcong	40.934327, −74.643549	*LL Alewife*	Juv-Adult	September 2015	ND	Boatmounted Umbrella net	ND	ND	n = 65

**Table 2 viruses-16-00824-t002:** Tissues collected and investigation methods used during disease surveillance of river herring collected from sampled New Jersey locations, including the Maurice River (Maurice), Great Egg Harbor River (GEH), Delaware River (Delaware) and Lake Hopatcong (Hopatcong). Abbreviations: Landlocked (LL), SP adult (Spawning adult), YOY (young of year), Juv (Juvenile), AK (anterior kidney), PK (posterior kidney), GI (gastrointestinal tract).

Site Name	Species	Life Stage	Sampling Period	Investigation Method	Tissues Collected
Maurice	*Alewife*	Sp adult	April 2015	histology, viral cell culture, qPCR	AK, PK, spleen, gill
Maurice	*Blueback*	Sp adult	April 2015	histology, viral cell culture, qPCR	AK, PK, spleen, gill
Maurice	*Alewife*	Sp adult	March–April 2016	histology	AK, PK, spleen, liver, heart, GI
Maurice	*Alewife*	Sp adult	April–May 2018	histology, qPCR	AK, PK, spleen, liver, heart, GI
Maurice	*Blueback*	Sp adult	April–May 2018	histology, qPCR	AK, PK, spleen, liver, heart, GI
GEH	*Alewife*	Sp adult	April 2018	histology, qPCR	AK, PK, spleen, liver, heart, GI
GEH	*Alewife*	YOY	August 2015	viral cell culture, qPCR	AK, PK, spleen, gill, brain
GEH	*Blueback*	YOY	August 2015	viral cell culture, qPCR	AK, PK, spleen, gill, brain
Delaware	*Blueback*	YOY	August 2015	viral cell culture, qPCR	AK, PK, spleen, gill, brain
Lake Hopatcong	*LL Alewife*	Juv-Adult	September 2015	viral cell culture, qPCR	AK, PK, spleen, gill, brain

**Table 3 viruses-16-00824-t003:** Primer sequences and PCR protocols. For resequencing, a 25 µL reaction was conducted using a TaKaRa LA PCR™ Kit (TaKaRa Bio, Shiga, Japan) Version 2.1 consisting of 5 µL of 10× LA PCR Buffer II (Mg^2+^ free), 1 µL dNTP Mixture, 0.375 µL (10 µM) each of F (Primer ID 1 or 3) and R primers (Primer ID 2 or 4), 15.75 µL of nuclease-free water, 0.5 µL of TaKaRa LA Taq, and 1 µL DNA template. For qPCR, a 20 µL reaction was conducted using 0.7 µL (10 µM) each of F, R, and Probes (Primer ID 5–7), 10 µL TaqMan™ Universal PCR Master Mix (ThermoFisher Scientific, Waltham, MA, USA), 3.9 µL nuclease-free water, and 1 µL DNA template.

Primer ID	Primer Name	Sequence (5′ → 3′)	Purpose	PCR Conditions
Resequencing PCR				
1	ApHBV802 F	5′-TTACAGCTACAGGGCATCAA-3′	Resequencing reaction 1 F	30 cycles of 10 s at 98 °C, 60 s at 60 °C, and 5 min at 68 °C. Final hold at 4 °C
2	ApHBV2914 R	5′-CAAAACAGCAGATGCGATAC-3′	Resequencing reaction 1 R	
3	ApHBVGapF	5′-CACGCGGTTTAGTGCTAACG-3′	Resequencing reaction 2 F	
4	ApHBVGapR	5′-GCAAGCCCAGTGAAACCAAG-3′	Resequencing reaction 2 R	
Quantitative PCR				
5	ApHBV438_191F	5′-CTTGGTTTCACTGGGCTTG-3′	qPCR F Primer	2 min at 50 °C, 10 min at 95 °C, followed by 40 cycles of 15 s at 95 °C, 60 s at 60 °C, and 15 s at 95 °C
6	ApHBV_555R	5′-AGAATGGGAGCATTCGGTGG-3′	qPCR R Primer	
7	ApHBV probe	5′-/56-FAM/CTGGACGCA/ZEN/GACCCCAGCAG/3IABkFQ/-3′	qPCR Probe	

**Table 4 viruses-16-00824-t004:** Polymerase protein pairwise sequence alignment identity (percentage) comparison to sample M10. Accession # indicate NCBI accession numbers, asterisks denote sequences obtained from NCBI SRA experiments in lieu of accession numbers. Novel alewife hepatitis B virus accession numbers are identified with bold and italics.

Accession #	Host	Tissue Source	Virus	% Similarity (M10)
WLK26110	*Alosa pseudoharengus*	Viscera	Alewife hepatitis B virus	-
WLK26114	*Alosa pseudoharengus*	Liver	Alewife hepatitis B virus	99.1
WLN26306	*Julidochromis dickfeldi*	Lower pharengeal jaw	Lamprologini metahepadnavirus	52.6
WLN26302	*Cyathopharynx furcifer*	Lower pharengeal jaw	Ectodini metahepadnavirus 1	51.1
SRX229523 *	*Astyanax mexicanus*	Eyes-surface	Mexican tetra hepadnavirus	51.0
WLN26312	*Ophthalmotilapia ventralis*	Lower pharengeal jaw	Ectodini metahepadnavirus 1	50.8
ERX674915 *	*Astatotilapia* sp.	Unknown	Astatotilapia metahepadnavirus	50.6
WLN26311	*Cyathopharynx furcifer*	Lower pharengeal jaw	Ectodini metahepadnavirus 3	50.5
WLN26301	*Simochromis diagramma*	Lower pharengeal jaw	Simochromis diagramma metahepadnavirus	50.4
WLN26310	*Xenotilapia* sp.	Lower pharengeal jaw	Xenotilapia metahepadnavirus	50.4
WLN26300	*Lamprologus lemairii*	lower pharyngeal jaw	Lamprologini metahepadnavirus	50.3
WLN26303	*Aulonocranus dewindti*	Lower pharengeal jaw	Ectodini metahepadnavirus 1	49.8
WLN26313	*Callochromis pleurospilus*	Gill	Ectodini metahepadnavirus 2	48.8
WLN26304	*Enantiopus melanogenys*	Lower pharengeal jaw	Ectodini metahepadnavirus 2	47.7
WLN26305	*Ophthalmotilapia ventralis*	Gill	Ectodini metahepadnavirus 3	47.0
SRX145766 *	*Boulengerochromis microlepis*	Lower pharengeal jaw	Boulengerochromis microlepis metahepadnavirus	47.0
AYU58612	*Hyporhamphus australis*	Gills	Eastern sea garfish hepatitis B virus	43.7
YP_009259541	*Lepomis macrochirus*	Lip	Bluegill hepatitis B virus	43.1
AYU58611	*Pagrus auratus*	Liver	Australasian snapper hepatitis B virus	34.4
KR229754	*Catostomus commersonii*	Liver	White Sucker hepatitis B virus	33.3
SRX1037831 *	*Oncorhynchus kisutch*	Kidney	Coho Salmon parahepadnavirus	31.0

## Data Availability

Any use of trade, firm or product names is for descriptive purposes only and does not imply endorsement by the U.S. Government. The USDA is an equal opportunity employer and provider. The original data presented in the study are openly available in GenBank at accession numbers OQ859058 (isolate M10) and OQ859059 (isolate MR4). At the time of publication, fish annual survey data were not publicly available from the New Jersey Division of Fish and Wildlife. Figure 1 Map Metadata: State layers downloaded from the US Census Bureau (USCB, 2018). United States Census Bureau, 2018. 2018 TIGER/Line Shapefiles (machine readable data files). U.S. Department of Com-merce. https://www.census.gov/cgi-bin/geo/shapefiles/index.php (accessed on 12 Decemeber 2022) Streamlines available from the NHDPlus Dataset Version 2.1 (EPA and USGS, 2012). U.S. Envi-ronmental Protection Agency & the U.S. Geological Survey. (2012). National Hydrography Dataset Plus—NHDPlus Version 2.1. http://www.horizon-systems.com/nhdplus/nhdplusv2_home.php.

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
