# Peer review of "Discovery and Genomic Characterization of a Novel Hepadnavirus from Asymptomatic Anadromous Alewife (Alosa pseudoharengus)"

_viruses, 2024, doi:10.3390/v16060824_

Round 1
Reviewer 1 Report
Comments and Suggestions for Authors
This manuscript mainly reports the full genome of a novel hepadnavirus from clinically normal alewives. The authors also develop a qPCR for hepadnavirus detection, and compare the viral genome sequence obtained in 2018 with the first virus. However, CPE, pathological change and hepadnavirus-like were not observed. While, observation of virus particles in the infected tissue is important for the virus identification. It is suggested to provide the micrographs of virus particles in the positive tissues observed by TEM. In addition, although histopathologic evaluation did not detect lesions, the histopathologic images could be added in the manuscript.
Reviewer 2 Report
Comments and Suggestions for Authors
The paper by Iwanowicz, Raines and colleagues is interesting and well written, mainly described the discovery and characterization of a novel hepadnavirus from Alewife. The study is significant and presents some valuable findings related to this novel virus. While the data presented supports the authors conclusions, the manuscript would be improved if the following concerns were addressed:
1. The study used high-throughput sequencing to analyze fish samples, revealing the presence of the ApHBV. However, only DNA nucleic acid was sequenced to identify potential DNA viruses in the samples. Have the authors considered employing RNA-Seq technology, possibly coupled with reverse transcription, to comprehensively detect both RNA and DNA viruses in the tissue samples? This approach would provide a more comprehensive assessment of the viral landscape within the herring population, allowing for a more thorough understanding of the viruses affecting the species.
2. Table 1, is this should be “Sample number” instead of “Sample size”? Please make sure this title is the correct description.
3. Consistency in table formatting enhances readability and professionalism. In the study, the authors' three table formats are inconsistent, so it is suggested to adjust them to a consistent table format for further published.
4. In Table 2, the author collected different type of internal tissues from different fish or the same kind of fish for relevant qPCR detection, cell culture or pathological tissue observation. I am confused about why the author did not use the same type of tissues (for example, only spleen and kidney were collected from each kind of fish for comparison) to conduct comparative research between the same kind of fish or different kinds of fish. Please explain it.
5. To enhance readability and academic clarity, it is recommended to interchange the descriptions of ORFZ and ORFY within their respective sections. Aligning these descriptions with the translation direction of the virus gene will provide readers with a more coherent understanding of the gene sequence and its functional implications.
6. For high-throughput sequencing, the specific fish species, tissue types, and age groups utilized are not explicitly mentioned. Clarifying these details would provide a clearer understanding of the experimental design and the relevance of the findings to specific populations or conditions.
Reviewer 3 Report
Comments and Suggestions for Authors
Discovery and Genomic Characterization of a Novel Hepadnavirus from Asymptomatic Anadromous Alewife (Alosa pseudoharengus), by Raines et al.
This is a well-designed study, and a well-organized and well-written manuscript. It has been a pleasure to review it, and I have only a few minor suggestions and must propose the inclusion of certain information:
- A section in M&M should be included to describe the qPCR assay, how the primers were selected, was it SYBR-GREEN or TaqMan, how its reliability was assessed, which was the standard (what target control, which hepadna strain?). Otherwise, too many things to leave to the reader's imagination.
- In addition, the readers might feel curious about why after the negative results from CC, histology and EM the authors decide to apply qPCR with hepadnavirus (which one?) as the target: There was any clue to suspect the etiology?
- I personally do not like to include in Results sentences more appropriate for Discusion, like those in lines 285 (… as observed in the …”) and 307-309. But, if the authors deem it really necessary in that way, I will not insist.
- Among the supplementary material, there is a excel file with the same tables shown in the manuscript. Is this material really needed? I do not think so.
Minnor corrections
1/ Line 69.- The common name of a group of viruses must be written in lower case: change “Orthohepadnaviruses” by “orthohepadnaviruses”.
2/ Lines 82-83.- Instead …”in a fish presenting with a tumor …” shouldn’t it be “… in a fish showing a tumor …”
3/ Line 156.- “GPS coordinates” are actually not shown. Do the authors mean “black dots correspond to sampling sites”?
4/ Line 193.- Delete the “,” in the sentence “A draft linear, genome”
5/ Lines 198 …- Please, indicate brand for the Equphi kit. For instance, should it be Thermo Scisntific?
6/ Line 265.- “N identifies the total …” should be “n identifies the …”
7/ Line 296.- What software was used for the protein characteristics prediction. Shouldn’t it be indicated in M&M?
8/ Line 332.- Insert “of” before “hepadnavirus”
9/Lines 332-334.- The sentence is too long. Please correct it to make it easier to follow.
10/ Line 343.- Change “… to the most similar …” by “… to be most similar …”
11/ Figure 6 and line 356-357.- Help to better identify the alewife strain in the figure, for instance using a larger font size in addition to bold and italics.
12/ Line 366.- Shouldn’t it be “… groups most closely to …” instead of “… groups most similarly to …”?
13/ Line 395.- Shouldn’t it be “… using samples of lower pharyngeal …” instead of “… using sampled lower pharyngeal …”?
14/ Line 400.- Change “>30” by “more than 30”
15/ Line 419.- Change “<1000” by “below 1000” or “less than” or …
16/ Line 424.- Delete the “,” in the sentence “… species or evaluation of …”
17/ Line 437.- Shouldn’t it be “A lack of detections among of these type of viruses may suggest their absence” instead of “A lack of detections among of this virus may suggest an absence”?
18/ Line 475.- Delete the “,” in the sentence
Round 2
Reviewer 1 Report
Comments and Suggestions for Authors
There is no more comment.